# Isolation and Identification of Bacteria with Surface and Antibacterial Activity from the Gut of Mediterranean Grey Mullets

**DOI:** 10.3390/microorganisms9122555

**Published:** 2021-12-10

**Authors:** Rosanna Floris, Gabriele Sanna, Laura Mura, Myriam Fiori, Jacopo Culurgioni, Riccardo Diciotti, Carmen Rizzo, Angelina Lo Giudice, Pasqualina Laganà, Nicola Fois

**Affiliations:** 1AGRIS-Sardegna, Agricultural Research Agency of Sardinia, Bonassai, 07100 Sassari, Italy; gabsanna@agrisricerca.it (G.S.); lmura@agrisricerca.it (L.M.); mfiori@agrisricerca.it (M.F.); jculurgioni@agrisricerca.it (J.C.); rdiciotti@agrisricerca.it (R.D.); nfois@agrisricerca.it (N.F.); 2Stazione Zoologica Anton Dohrn-Ecosustainable Marine Biotechnology Department, Sicily Marine Centre, Villa Pace, Contrada Porticatello 29, 98167 Messina, Italy; carmen.rizzo@szn.it; 3Institute of Polar Sciences, National Research Council (ISP-CNR), 98122 Messina, Italy; angelina.logiudice@cnr.it; 4Department of Biomedical and Dental Sciences and Morphofunctional Imaging, University of Messina, Torre Biologica 3p, AOU ‘G. Martino, Via C. Valeria, s.n.c., 98125 Messina, Italy; plagana@unime.it

**Keywords:** intestinal microflora, fish gut, biosurfactants, grey mullets, natural antibiotics

## Abstract

Fish gut represents a peculiar ecological niche where bacteria can transit and reside to play vital roles by producing bio-compounds with nutritional, immunomodulatory and other functions. This complex microbial ecosystem reflects several factors (environment, feeding regimen, fish species, etc.). The objective of the present study was the identification of intestinal microbial strains able to produce molecules called biosurfactants (BSs), which were tested for surface and antibacterial activity in order to select a group of probiotic bacteria for aquaculture use. Forty-two bacterial isolates from the digestive tracts of twenty Mediterranean grey mullets were screened for testing emulsifying (E-24), surface and antibiotic activities. Fifty percent of bacteria, ascribed to *Pseudomonas aeruginosa*, *Pseudomonas* sp., *P. putida* and *P. anguilliseptica, P. stutzeri, P. protegens* and *Enterobacter ludwigii* were found to be surfactant producers. Of the tested strains, 26.6% exhibited an antibacterial activity against *Staphylococcus aureus* (10.0 ± 0.0–14.5 ± 0.7 mm inhibition zone), and among them, 23.3% of isolates also showed inhibitory activity vs. *Proteus mirabilis* (10.0 ± 0.0–18.5 ± 0.7 mm inhibition zone) and 6.6% vs. *Klebsiella pneumoniae* (11.5 ± 0.7–17.5 ± 0.7 mm inhibition zone). According to preliminary chemical analysis, the bioactive compounds are suggested to be ascribed to the class of glycolipids. This works indicated that fish gut is a source of bioactive compounds which deserves to be explored.

## 1. Introduction

The study of biodiversity for exploring new biological sources is considered a suitable approach in the *bioprospecting* field for the discovery of new bioactive molecules in nature [1]. Marine environments represent some of the most interesting places for the isolation of new metabolites, due to their unique and variable physical surroundings which induce the producer microorganisms to develop metabolic and physiological capabilities for adapting to diverse habitats covering a wide range of thermal, pressure, salinity, pH and nutrient conditions [2,3,4]. Indeed, the identification and production of broad-spectrum activity microbial compounds have been obtained from different aquatic ecosystems and matrices [5,6,7]. However, to date, most of the marine microbial world is still scarcely explored, and the interest in novel compounds remains the main driver of different research projects.

Fish are generally valued for their qualities as goods, in the form of food protein, fishmeal, fish oil, for aquaculture and in the pharmaceutical industry for the production of medicine [3]. In particular, fish gut was found to be a peculiar ecological niche where bacteria transit and reside to play different vital roles (protective barriers against pathogens, promotion of fish immunity, fish nutrition, and so on) [8]. To date, different studies have described fish intestinal microbiota as a reflection of the environment and a variety of other factors (genotype, physiological status, fish behaviour, feeding habit); however, these works concerned the composition of the microbial community, the isolation, identification of microorganisms and the possible use of single bacterial culture or consortia of strains to promote fish growth and health [8,9,10,11,12,13]. Recently, the importance of marine fish gut microbiota has been highlighted in terms of the production of biofilm barriers formed by extracellular polymeric substances able to protect the host organism and protect it against pathogens [14]. Previous studies on the production of metabolites with surface activities, called biosurfactants (BSs), from fish intestinal microbial content, have highlighted the importance of the intestinal tract of fish in the search for new molecules [6]. The BS-producing microorganisms are ubiquitous, inhabiting both water (seawater, freshwater and ground water) and land (soil, sediment and sludge), as well as environments characterized by extreme conditions of pH, temperature or salinity (e.g., hyper saline sites and oil reservoirs) [15]. BSs have been commonly studied for their bioremediation properties and for antibacterial, antifungal and antiviral activities [16], and they have been found in polluted environments [17] as well as in different biological matrices [18]. Recently, the immunomodulatory role of microbial BSs has been highlighted by Giri et al. [19], and the antibacterial activity of fish gut, associated with bacteriocinogenic bacteria, was detected by Mukherjee et al. [20]. Nevertheless, the gastrointestinal microflora of fish remains a little-explored subject of basic and applied research.

Mugilidae, commonly known as grey mullet, comprise the highest number of fish species, and are among the most ubiquitous teleost families in the coastal waters of the world [21]. Grey mullets have been described as mud-eaters, as well as detritus, deposit and interface feeders; their gastrointestinal tracts, 1.5–4.6 times longer than the total body length, are arranged in several convolutions and wrapped in a peritoneal connective tissue which acts as a fat storage [22]. For these reasons, these fish constitute an interesting source of bacteria with peculiar biochemical characteristics due to the relative importance of detritus and algae in the diet. As regards Mediterranean Mugilidae, they are the main representative fish species in Sardinian estuaries and lagoons. They are appreciated in the food market, and *Mugil cephalus* roe, called “bottarga”, represent an added value product and a highly prized delicacy in the southern Mediterranean [23]. Different studies have focused on the genetics [24], biogeography, distribution [25] and potential probiotics of these fish [26,27,28], but to the best of our knowledge, no works on the production of bioactive compounds from the intestinal microflora of the Mugilidae have been published. The aim of this study was to identify and investigate BS microbial producers isolated from the gut of different Mediterranean mullets, *Mugil cephalus*, *Chelon ramada*, *Chelon labrosus* and *Chelon saliens*, in order to assess them as a source of natural added-value bioactive compounds to be selected for aquaculture field and bioremediation. 

## 2. Materials and Methods

### 2.1. Study Area, Sampling and Microbiological Analysis

Twenty wild-caught mullets, *Mugil cephalus* (*n* = 4), *Chelon ramada* (*n* = 6), *Chelon labrosus* (*n* = 8) and *Chelon saliens* (*n* = 2), destined for the local food market, were captured by professional fisheries on 27 September 2018 (autumn) and 10 February 2019 (winter). The study area was Santa Giusta lagoon, an 8 km^2^ area with a mean depth of approximately 1 m (Sardinia, Italy; coordinates: Lat 39°52′N, Long 8°35′E). Water temperature (Temp: 12.5–28.0 °C), salinity (Sal: 28.0–44.0 ppm) and dissolved oxygen (DO: 7.0–11.5 mg L^−1^) were measured in situ using a YSI 6600 v2 (YSI Inc., Yellow Springs, OH, USA) multi-parameter probe. This lagoon is peculiar for its recurring ecological instability due to different anthropogenic impacts since 2000 [29]. The fish (average size and weight, 30.1 ± 5.8 cm and 277.4 ± 183 g, respectively) were transported inside a refrigerated bag to the Bonassai laboratory within 6–8 h, weighed using a scale (d = 0.01 g) and measured in length (d = 0,1 cm). The entire intestine (mean weight 14.0 ± 6.0 g) was aseptically removed from each fish, diluted (10% *w*/*v*) in saline solution (0.9% NaCl) and homogenized in plastic bags by a Stomacher^®^ 400 (FermionX Ltd., Worthing, UK) at room temperature. Samples were made by mixing the guts of two individuals of each species in order to obtain five samples for each date, up to a total of ten samples. Serial dilutions of the homogenate were prepared, and 100 µL of each dilution were spread on Marine agar (MA, Himedia, Mumbay, India) plates in duplicate and incubated at 30 °C for 48 h, for the enumeration of heterotrophic marine bacteria. 

Bacterial colonies were randomly isolated and streaked onto fresh medium four times to obtain pure cultures. The purified isolates were stored at −80 °C in a 15% (*v*/*v*) glycerol-Nutrient broth (NB, Conda Pronadisa, Madrid, Spain) solution. The strains were assayed for the BS production, as follows. 

### 2.2. Screening of Bacteria for Biosurfactant Production 

#### 2.2.1. Detection of the BSs in the Culture Broth

Each isolate was tested for BS production using a battery of screening tests, performed in three independent experiments as replicates. A loopful of each strain from well-grown MA plates was used to inoculate the bacterial cultures in 100 mL sterile Erlenmeyer flasks containing 50 mL of Bushnell–Haas medium (BH, Himedia) supplemented with sunflower oil (2% *v*/*v*). The cultures were incubated at 25–28 °C under shaking for 48–96 h, and growth was monitored after 24, 48, 72 and 96 h by measuring the optical density at 600 nm (OD_600 nm_) with a spectrophotometer (Cary 1E UV-Visible Spectrophotometer, Varian Instruments, Sugarland, TX, USA); pH values were also registered. Bacterial cultures were screened for BS production using standard screening tests, performed as described in the following sections. Sodium dodecyl sulphate (SDS) and Tween 80 were used as positive controls, whereas distilled water and BH medium plus sunflower oil were used as negative controls.

Bacterial pure cultures were tested during the late stationary growth phase by means of the emulsification index (E-24) and the drop-collapse assay.

#### 2.2.2. Emulsification Index (E-24)

The emulsifying capacity was evaluated using the emulsification index detection (E-24) according to [30]. An equal volume of kerosene and culture broth (2 mL) was vortexed at high speed for 2 min in test tubes and allowed to stand for 24 h. The E-24 index is given as the height of the emulsified layer (cm) divided by the total height of the liquid column (cm) and expressed as a percentage. 

#### 2.2.3. Drop-Collapse Assay 

The tests were performed using the polystyrene lids of a 96-microtiter 12.7-by 8.5-cm BRANDplates^®^ (Greiner Bio-One GmbH, Frickenhausen, Germany) according to [31], with some modifications. A 1.8 µL aliquot of diesel oil was added to each lid’s well and equilibrated for 24 h at room temperature. A 5 µL amount of the culture broth was then added to the surface of the oil previously placed in the centre of each well. The shape of the drop on the oil surface was inspected after 1 min. BS-producing cultures, which collapsed, giving flat drops, were scored as positive (+), while those which gave rounded drops and remained beaded were scored as negative (–). 

#### 2.2.4. Surface Tension Measurement

All isolates that proved to be positive in the previous screenings were tested with the Wilhelmy Plate method. The surface tension was determined on the cell-free supernatant of the bacterial cultures after centrifugation at 10,000 rpm for 20 min at 4 °C. Supernatants were stored at −20 °C and successively measured with a digital tensiometer using the Wilhelmy Plate method according to [18]. A surface tension lower than 40 mN·m^−1^ was considered as an index of BS production.

### 2.3. Biosurfactant Extraction and Thin Layer Chromatography (TLC)

The BSs-producing bacterial cultures were centrifuged at 10,000 rpm for 20 min and the supernatants characterized by thin-layer chromatography (TLC) according to [32]. The BSs were extracted from 5 mL of supernatants. The pH of the supernatants was adjusted to 2.0 with 1 N HCl and left at 4 °C overnight [33]. The extraction of the BSs was carried out twice, adding an equal volume of chloroform:methanol (2:1, *v*/*v*). The mixture was vigorously shaken for 1 min and allowed to stand until the phase separation. The organic layer (lower phase) was retained and concentrated under vacuum using a rotary evaporator at 40 °C. Successively, the extracts were weighted to acquire the amount of the crude yield, re-suspended in 200 µL chloroform:methanol (2:1, *v*/*v*) mixture and analysed by thin-layer chromatography (TLC) on silica gel plate (© Millipore Corporation, Burlington, Massachusetts, USA). The solvent system used was chloroform:methanol:acetic acid:water (65:15:1:1, *v*/*v*/*v*/*v*), and for detecting the less polar compounds, hexane:ether:acetic acid (70:30:2, *v*/*v*/*v*) was used. The TLC run lasted for approximately 90 min. The TLC plates were stained in two different solutions. The sugar moieties were identified by staining the plates with anisaldehyde (Sigma-Aldrich, Burlington, MA, USA):glacial acetic acid:sulphuric acid:ethanol (0.15:1:2:37, *w*/*v*/*v*/*v*), while, the fatty acid moieties were stained with an ammonium molybdate (Sigma-Aldrich, Burlington, MA, USA):cerium sulphate (Sigma-Aldrich, Burlington, MA, USA):sulphuric acid:water (3:0.5:8.5:23.45, *w*/*w*/*v*/*v*) solution. The colour of the spot on the plate was developed by heating inside an oven at 150 °C. The TLC patterns of the extracts were compared with those of three different standards for the identification of the BSs: Sophorolipids (S) (Sopholiance, Reims, France), a Trehalose Lipid Tetraester (crude extract) (T) (Karlsruhe Institute of Technology, Karlsruhe, Germany), Phospholipid mixture for HPLC (Supelco, Bellefonte, Pennsylvania) and a mix of Rhamnolipids R-95 (R) (Sigma-Aldrich, Burlington, MA, USA). Each retardation factor (Rf) was calculated by dividing the distance of the considered TLC fraction run in the TLC plate from the origin by the distance of the solvent from the same origin.

### 2.4. Bacterial Identification 

Bacterial isolates were identified by partial 16S rRNA gene sequencing. Bacterial cell preparation for DNA extraction were performed using a Qiagen kit (DNeasy^®^ Blood & Tissue Kit, Hilden, Germany). Universal primers designed to amplify approximately 1300 bp of *Escherichia coli* 16S rRNA gene were used [34]. The sequences were: forward primer 63f (50 -CAG GCC TAA CAC ATG CAA GTC-30) and reverse primer 1387r (50 -GGG CGG WGT GTA CAA GGC-30). PCR mixture contained from 50 to 100 ng DNA template, 1 µL of each primer (50 pmol µL^−1^) (Sigma Genosys, The Woodlands, Texas, USA) and 20 µL of ready-to-use PCR master mix containing *Taq* polymerase (MegaMix 2MM-5, µ Microzone Limited, Stourbridge, UK), to give a total reaction of 25 µL. The PCR conditions were: 30 cycles of denaturation at 94 °C for 1 min, annealing at 58 °C for 1 min and elongation at 72 °C for 2 min, with a final elongation at 72 °C for 10 min. Purification of the amplicons for sequence study was carried out as described in the Qiagen (QIAquick^®^ PCR Purification Kit, Hilden, Germany) protocol. Partial sequences were determined by BMR Genomics s.r.l (Padova, Italy). The sequencing results were submitted for homology searches by BLAST (Basic Logical Alignment Search Tool) [35] after unreliable sequences at the 3′ and 5′ ends were removed using the software Chromas, version 1.43 (Griffin University, Brisbane, Qld, Australia). The NCBI GenBank nucleotide database (National Center for Biotechnology Information, http://www.ncbi.nml.nih.gov accessed on October 2021) was used for sequence pairing. The identities were determined on the highest score basis. Nucleotide sequences were deposited in the NCBI GenBank database under the accession numbers MW369461- MW369487 [11] and OK342256-OK342267 (This study). A phylogenetic tree was reconstructed using the 16S rRNA gene sequences obtained in the study and the reference strains ENA AF094713 *P. aeruginosa* ATCC 10145^T^, ENA X60410 *Aeromonas media* ATCC 339007^T^ and ENA AJ853891 *Enterobacter ludwigii* DSMZ 16688^T^. As an out-group, the strain NR074804 *Cellvibrio japonicus* strain Ueda 107 was utilized for the analysis [36]. A Clustal W Multiple alignment was obtained by MEGA X [37]. The final dataset was included using 820 bp positions. The best fit DNA evolution model selected on the IQTREE webserver [38] was K2P G4 [39]. A phylogenetic tree was inferred by the Maximum Likelihood algorithm [40,41] in IQTREE with default parameters. The bootstrap test (1000 replicates) [42] was used to evaluate the robustness of the tree topology.

### 2.5. BS Antibacterial Activity

#### 2.5.1. Bacterial Pathogens

The antibacterial activity was tested against the following bacterial pathogens: *Pseudomonas aeruginosa* H1628, *Staphylococcus aureus* H1670, *Klebsiella pneumoniae* H1637, *Proteus mirabilis* H1643 and *Aeromonas hydrophila* H1563. The strains were previously isolated from human clinical specimens and identified to the species level by API 20 E, API 20 NE and API STAPH profiles (bioMérieux, Marcy l’Etoile, France). Lab strains are maintained at −20 °C in Tryptone Soy Broth (TSB, Difco) supplemented with 15% glycerol. 

#### 2.5.2. Antibacterial Activity

The inhibitory activity was tested on cell-free supernatants (CFSs) and crude extracts (CEs) using the standard disk diffusion method (DDM) (Kirby Bauer test), as accepted by the National Committee for Clinical Laboratory Standards (NCCLS 2000). Details are described below.

Cell-Free Supernatants

CFSs were obtained through centrifugation at 10,000 rpm at 4 °C for 20 min of cell culture aliquots, and filter-sterilized on nitrocellulose membranes (pore diameter 0.22 µm). Each CFS was ten-fold concentrated prior to testing using a concentrator (Concentrator 5301, Eppendorf AG, Hamburg, Germany), as described by [43]. Bacterial pathogens were suspended in 3 mL of a saline solution (NaCl 0.9%, *w*/*v*) in order to achieve a turbidity of McFarland 0.5 standard (containing around 1.5 × 10^8^ cells/mL), and the suspensions were spread-plated on plates of TSA supplemented with 1% (*w*/*v*) NaCl (TSA1), in triplicates. Aliquots (60 µL) of each CFS were used to soak sterile cellulose discs (6 mm diameter), which were laid on the medium surface previously inoculated with pathogenic strains. Distilled water (20 µL) was used to soak sterile disks as a negative control, while commercially available disks (6 mm in diameter, Oxoid) containing chloramphenicol (30 µg), amoxicillin (30 µg) and gentamycin CN30 (30 µg) were used as a positive control. The plates were incubated for 24 h at 37 °C.

Crude Extracts

The extraction of CE was performed on 50 mL aliquots of each CFS (obtained as described in the section above). Firstly, CFSs were acidified with phosphoric acid (85%, *v*/*v*), and bioactive molecules were extracted twice in ethyl acetate (cell-free supernatant: ethyl acetate ratio 1:1.25, *v*/*v*). Ethyl acetate was totally evaporated at room temperature and extracts were collected [7]. Based on the total amount, each CE was dissolved in a proper volume of ethyl acetate in order to obtain 6 mg of extract in a final volume of 20 µL. After complete solvent evaporation, the disks were placed onto TSA1 plates inoculated with the target pathogens. Disks soaked with ethyl acetate and submitted to evaporation were used as negative controls, while positive controls were performed as for CFSs tests. Plates were incubated overnight at 37 °C. The diameter of complete inhibition zones was measured, and means and standard deviations (*n* = 3) were calculated. The results were codified as weak activity for inhibition zone lower than 8 mm [44] European pharmacopoeia.

## 3. Results

### 3.1. Enumeration of Bacteria and Colony Isolation 

Bacterial counts on MA medium showed values of heterotrophic marine bacteria from 10 × 10^3^ to 10 × 10^4^ colony forming units (CFU) in autumn and from 12 × 10^4^ to 40 × 10^4^ CFU in winter. Forty-two bacterial colonies were isolated from different mullet species (strains 1–26 from fish sampled in autumn and strains 28–56 in winter). 

### 3.2. Screening of Bacteria for BS Production

In this study, the intestinal bacterial strains isolated from mullet grey fish showed a diversity in the bioactive performances. Thirty-three out of 42 strains were able to utilize sunflower oil for their growth as the sole energy and carbon source in BH medium at 25–28 °C after 72 h. The application of different screening methods allowed the selection of a “group” of intestinal strains as surfactant producers in three independent experiments. The drop-collapse method was used as a first screening test for identifying the “bioactive” microbes. Table 1 shows the results of all the used screening tests. By means of the drop-collapse assay, a surface activity of different intensity was detected: eight *Pseudomonas aeruginosa* (i.e., strains 1, 3, 5, 6, 9, 12, 13 and 15) gave a strong positive score (+++), two *Pseudomonas aeruginosa* (i.e., strains 8 and 26) were scored as pretty good surface active cultures (++), eight intestinal strains were ascribed to *Pseudomonas* spp. (i.e., strains 10, 19, 22, 25, 41, 45, 51 and 56) and one isolate, *Enterobacter* sp. (strain 28), presented a discrete activity (+), while seven *Pseudomonas* spp. isolates (i.e., strains 17, 18, 20, 23, 24, 47 and 55), five strains ascribed to *Aeromonas* spp. (i.e., strains 11, 30, 35, 37 and 40) and two unidentified cultures showed a weak or absent activity, scored as (weak) or (-) (Table 1). The emulsifying cultures showed stable and compact emulsions with kerosene at the end of the exponential and/or during the stationary growth phase (OD 600 nm = 2.0–3.0) and remained stable over 1–2 months without any significant change in the index values. The values reached by means of the emulsification index (E-24), were from 0 to 70% after 72 h of incubation (Table 1). Generally, the greatest E-24(%) values were observed in the strains which had given the strongest positive score by the drop-collapse test, except for *P. aeruginosa* (strain 8), which showed a good activity by the drop-collapse assay but a null value of E-24(%), while strain 12 showed a weak E-24(%) and a high score by the drop-collapse test. 

On the other hand, the results of surface tension, measured by the Wilhelmy Plate method, showed a lower variability between strains, detecting values from 35 to 46 (mN·m^−1^) (Table 1). Figure 1 shows the emulsification indexes E-24(%) and the surface tension activity of the detected “bioactive” intestinal bacteria. 

The most interesting strains are represented by eight isolates (strain 1, 3, 5, 6, 9, 10, 13, 15), showing an emulsification index E-24(%) from 50.0 to 77.0% and a surface tension from 36.5 to 37.2 (mN·m^−1^). 

### 3.3. Bacterial Identification

Thirty strains, which showed bioactivity, were identified by 16S rRNA gene partial sequencing. Table 1 shows fish origin, phylogenetic affiliation and accession number of the studied intestinal strains. The gut microbiota of the mullets was ascribed to 14 different species: *Pseudomonas aeruginosa* (9 strains), *Pseudomonas* sp. (5 strains), *A. media* (3 strains), *P. protegens* (3 strains), *P. alcaligenes* (1 strain), *P. mendocina* (1 strain), *P. putida* (1 strain), *P. alcaliphila* (1 strain), *P. khazarica* (1 strain), *P. anguilliseptica* (1 strain), *P. stutzeri* (1 strain), *Aeromonas caviae* (1 strain), *A. taiwanensis* (1 strain), *Enterobacter ludwigii* (1 strain). Figure 2 shows the phylogenetic tree reconstructed using the 16S rRNA gene sequences. Different groups and subgroups were obtained with respect to the corresponding reference species type strains and among themselves. The most heterogeneous group was represented by *Pseudomonas* spp., while *Aeromonas* spp. and *Enterobacter* sp. formed well distinguished clusters. The outgroup strain NR074804 *Cellvibrio japonicus* strain Ueda 107 was separated from all the others. 

### 3.4. BSs Extracts and Thin Layer Chromatography (TLC)

Figure 3 shows the BSs yield extracts of representative intestinal bacteria. The BS producers which gave the highest yield extracts (values from 6–6.42 g L^−1^) were strains 6, 13, 15.

The chromatographic analyses (TLC) of the BSs extracted from intestinal bacterial supernatants showed two types of glycolipid compounds (Figure 4). The TLC silica gel glass plates stained by anisaldehyde (carbohydrates) (Figure 4a,c) and cerium sulphate (lipids) (Figure 4b,d) indicate a group of specific TLC fractions (retardation factor Rf = 0.42), which presumably represents the di-rhamnolipid structures, while a group of other fractions (Rf = 0.75) detects the mono-rhamnolipid molecules (Figure 4a,b). These defined compounds were characterized by the same Rf values as the rhamnolipid standard and were exhibited by bacterial strains 1, 3, 5 and 13 (Figure 4a,b). Another type of molecule was also detected by the TLC analyses but not separated using the first solvent system indicated above (strains 19 and 26, Figure 4a,b). These less polar compounds from strains 6, 18, 19, 22, 25 and 26 were separated using a less polar solvent system, as described above, and gave different profiles of the TLC fractions (from the bottom, Rf = 0.20, 0.30, 0.60) (Figure 4c,d).

### 3.5. Antibacterial Activities

CFSs exhibited inhibitory activity against the target strains *S. aureus* H1610 and *P. mirabilis* H1643, while no inhibition was evidenced against the target strains *P. aeruginosa* H1328, *K. pneumoniae* H1637 and *A. hydrophila* H1563 (Table 2). 

Of the tested CFSs, 26.6% exhibited antibacterial activity, with halos ≥10 mm compared to the target strain *S. aureus* H1610 (Figure 5a). Specifically, CFSs 1 and 56 (from *Pseudomonas aeruginosa* and *Pseudomonas* sp., respectively) recorded inhibitory halos of 13.5 ± 0.7 mm, while CFS 15 (from *Pseudomonas aeruginosa*) showed the highest inhibitory activity of 14 ± 0.0 mm. CFS 6 (*Pseudomonas aeruginosa*) showed a weak activity, indicated as a positive response. Of the CFSs, 50% resulted as active against the target strain *P. mirabilis* H1643, with seven CFSs exhibiting inhibition activity for halos ≥10 mm (*Pseudomonas aeruginosa* 1 and 3, *Pseudomonas* sp. 22, *Pseudomonas aeruginosa* 26, *Aeromonas media* 37 and CFSs from strains 12 and 16), four CFSs (CFSs from *Pseudomonas aeruginosa* 13, *Enterobacter ludwigii* 28, *Pseudomonas protegens* 47 and *Pseudomonas* sp. 56), with only weak activity showing inhibition halos ≤10 mm (8 ± 00 mm, 8 ± 00 mm, 9 ± 00 mm, 7 ± 00 mm, respectively). In this case, the highest inhibitory activity was shown by the CFS of *Pseudomonas aeruginosa* 3, with an inhibition halo of 18.5 ± 0.7 mm (Table 2 and Figure 5b).

The CEs evidenced antibacterial activity against more target strains, namely *S. aureus* H1610, *P. mirabilis* H1643, *K. pneumoniae* H1637 and *A. hydrophila* H1563, while no activity was recorded against *P. aeruginosa* H1628. Of the CEs, 76.7% and 66.7% were active against *S. aureus* H1610 and *K. pneumoniae* H1637, respectively, while 20% of the CEs showed inhibitory activity against *P. mirabilis* H1643 and *A. hydrophila* H1563 (Figure 5). Specifically, three CEs (*Pseudomonas aeruginosa* 9, 13 and 15) exhibited antibacterial activity, showing halos ≥10 mm against *S. aureus* H1610, with the highest inhibition for 9 (14.5 ± 0.7 mm) (Table 2). The rest of the tested CEs exhibited antibacterial activity, with halos ranging from 5.5 ± 0.7 to 9.5 ± 0.7 mm, and two weak responses have been recorded for CEs from *Pseudomonas* sp. 25 and *Pseudomonas anguilliseptica* 41 (Table 2 and Figure 4a). Six CEs, showing inhibitory activity against the target strain *P. mirabilis* H1643, evidenced inhibitory ≥10.00 ± 0.0 mm, ranging from 12.0 ± 0.7 mm (CEs *Pseudomonas aeruginosa* 6 and *Pseudomonas* sp. 25) to 16.0 ± 0.0 (CE *Pseudomonas alcaligenes* 10) (Table 2 and Figure 5b). Five CEs resulted active against *K. pneumoniae* H1637 with halos ≥10.0 mm, and four weak responses have been evidenced by CEs from *Pseudomonas alcaligenes* 10, *Aeromonas caviae* 11, *Enterobacter ludwigii* 28, *Aeromonas media* 40 and *Pseudomonas* sp. 56. The highest activity was obtained by CE of *Pseudomonas aeruginosa* 1 (17.5 ± 0.7 mm) (Table 2 and Figure 4c). Finally, the antibacterial activity against *A. hydrophila* was exhibited in all cases, with halos ≤10.0 mm, with two weak responses (CEs from *Aeromonas media* 40 and *Pseudomonas* sp. 19) and four inhibitory halos ranging from 6.5 ± 0.7 to 8.0 ± 0.0 mm (Table 2 and Figure 5d).

## 4. Discussion

Fish gut is a place where several metabolic activities take place and where bacteria can enter, reside or transit; for this reason, it deserves to be explored as a marine source of new added-value compounds. As an example, bacterial extracellular polymeric substances seem to play a pivotal role in the formation of complex biofilm architecture in marine fish gut, as demonstrated for the luminous bacteria isolated from the gastrointestinal tract of White Sea fish [14]. In the present study, bacterial strains from the intestinal tract of different mullet species from a brackish peculiar environment were assayed for the production of secondary metabolites such as biosurfactants with surface and antimicrobial compounds. The battery of tests applied led to the selection of 50% of strains able to produce molecules with a different spectrum of emulsifying and surface activities. However, culture conditions play a crucial role in the growth of a strain and in its production of a particular metabolite, as reported by many authors who tried to discover the optimum culture conditions and suitable hydrocarbon source to achieve the maximum yield of these compounds [32,45,46,47]. The study concerns bacterial isolates from different species of wild mullets sampled in two different seasons (autumn and winter). Noteworthy, the analyses of cell-free supernatants obtained from bacterial cultures isolated from fish in autumn provided evidence of their good emulsifying properties and their significant reduction in surface tension, while for most of the strains isolated from the gut of fish captured in winter, only a discrete surface tension activity was scored in their supernatants. The different behaviours of the bacterial culture bioactivities are interesting and can be ascribed to the environmental conditions, which varied a great deal in the lagoon during the two sampling periods. Indeed, the aquatic environment of capture is known for its recurring ecological instability due to different anthropogenic impacts [29]. These findings strengthen the idea of the relevant role of factors, such as temperature and salinity, on the variability of fish intestinal bacteria other than nutritional factors, host, fish habit and metabolic activity [11]. This is quite realistic because previous microbiological studies have indicated that the aqueous habitat influences fish microbial gut flora [10,11]. Moreover, the results reported in this work (surface tension values from 35.05 ± 0.4 mN·m^−1^ to 43.01 ± 1.2 mN·m^−1^) seem similar to those registered in other studies on *Pseudomonas* spp. (29–50 mN·m^−1^) [48,49,50,51]. In accordance with the present results, these authors stated that the production of the identified BSs (rhamnolipids) is probably connected to external conditions such as nutrient limitation or other environmental factors, thus playing a crucial role in modulating bacterial behaviour over microbial community life and environmental changes. However, the BS producer performances, observed in this study, were different from those found in other microbiological studies on the intestinal tract of *Sparus aurata* from different aquatic environments, during the winter season, where lower (E-24) values and stronger surface activities were registered for the majority of the tested intestinal strains [6]. Throughout this study, the interfacial activity and the emulsification capacity do not always correlate, and this is in line with what has been highlighted by several authors [6,15,18]. Overall, it is interesting to observe that the best performers for BS production mainly belonged to *Pseudomonas* spp. which represent ubiquitous bacteria in nature and were already found to be part of grey mullet and other fish species cultivable microflora [6,10,11,52]. The phylogenetic tree obtained using the 16S ribosomal RNA gene sequences indicated the presence of a heterogenous cluster of *Pseudomonas* spp. in the mullet intestinal bacterial flora, forming interesting groups and subgroups of strains that deserve to be further investigated for taxonomic purposes by other housekeeping genes [36]. Moreover, in this work, the preliminary chemical structure of BSs from a group of representative strains was analysed. On the basis of our results, it is suggested that these are ascribed to two classes of glycolipid molecules: rhamnolipids and less polar compounds. The glycolipid biosurfactants have recently gained special attention for their eco-friendly nature and high efficiency in biodegradation as well as other special activities such as pesticidal, antifungal and antibacterial activities [53]. Indeed, as reported in other studies, *Pseudomonas* genus is able to synthesize the BSs of a diverse chemical nature, and the more widely studied ones are low molecular weight compounds called rhamnolipids [48,54]. Bacterial rhamnolipids biosynthesis was first elucidated in the Gram-negative opportunistic pathogen *P. aeruginosa*, which can synthesize a range of rhamnolipid congeners (approximately 60), di-rhamnolipids (the most abundant) and mono rhamnolipids. They present low toxicity and high biodegradability and are naturally produced at different concentrations by other *Pseudomonas* spp., such as *P. fluorescens*, *P. chlororaphis*, *P. putida* and *P. mendocina* [51,55,56], though their level of production is low compared to *P. aeruginosa* strains [57]. However, *Pseudomonas* strains isolated from the gut of gilthead seabream from different Sardinian aquatic environments also resulted in the production of glycolipid compounds, although showing slight differences in the TLC profiles, with respect to this study [6]. Furthermore, it is important to highlight that there is a great variability in the bioactive performances of this type of molecules, and there are marked differences between biosurfactants and bioemulsifiers. Although both BS types can efficiently emulsify two immiscible liquids, bioemulsifiers are said to possess only emulsifying activity and not surface activity [58]. In any case, from a practical point of view, bacteria producing surface-active compounds, such as rhamnolipids, are thought to solubilise insoluble substrates such as hydrocarbons and to promote the uptake and the biodegradation of poorly soluble substrates, enhancing their bioavailability and subsequent metabolism [55]. Noteworthy, these natural compounds also act as immune modulators, virulence factors and antimicrobial agents and are involved in surface mobility and bacterial biofilm development [55]. Our experiments also referred to the antibacterial activity, another aspect of BS-producing strains which has important practical implications. In this study, CFSs and CEs showed different inhibitory activity toward target strains. Indeed, while CFSs resulted active only against two targets, the extracts resulted active, to a different extent, against all target strains. Interestingly, in some cases, the inhibitory activities achieved values similar or equal to those of the positive control. This is the case for many CFSs and CEs, which showed inhibition halos equal to or higher than those obtained using Gentamycin CN30 against the target *P. mirabilis* H1643. This is also true for the inhibition exhibited by both CFSs and CEs against the target *K. pneumonia* H1637. In particular, four *Pseudomonas* isolates (strains 1,6,13,15) from *Chelon ramada* were found to be the most effective, showing E-24 > 50%, a vigorous collapse in the drop collapse-test and antibacterial activities against diverse Gram-negative and Gram-positive pathogens, except for *P. aeruginosa* H1628. As reported in previous similar studies, extracts and supernatants showed differentiated activity [6]. Thus, the search for new molecules with antibacterial function finds pivotal potential application in the aquaculture field, which is seriously threatened by the spread of infectious diseases and, more seriously, by antibiotic-resistant bacteria caused by the excessive use of drugs [59]. Aquaculture production is projected to rise from 40 million tonnes by 2008 to 82 tonnes in 2050. Moreover, by 2030, farm-raised fish would account for nearly two-thirds of the world’s seafood intake, according to estimates by the United Nations Food & Agriculture Organization (FAO, 2010) [60]. The excessive use of antibiotics and the problems linked to this in aquaculture have been questioned by [61], who claim that the improved rearing methods may lead to the existence of antibiotic residues in seafood, with the consequences of destructing the immune system of the host. Recently, new scientific strategies are moving towards the supplementation of fish diets with several additives (i.e., probiotics, prebiotics, immunostimulants, vaccines) which could improve animal survival and wellness [62]. In this panorama, the development of a sustainable aquaculture industry is challenged by the limited availability of natural resources as well as the impact of the industry on the environment [63]. 

For all these reasons, the identification of bacteria able to produce natural compounds as suitable alternative to common antibiotics that reduce intestinal pathogens in animals and humans is of great importance and can increase the amount of information on the possible influence of intestinal bacteria on the health/well-being of fish. Consequently, the attention of fish farming practices should focus on this topic because their aim is to produce in large quantities while respecting the environment and animal welfare, in accordance with strict European rules on microbiological criteria for food market safety (Reg. ECN°2073/2005). Besides, the obtained results have important economic implications for an aquacultured species such as *Mugil cephalus*, which is highly appreciated for its eggs, processed to obtain seafood which is known by different names, such as Avgotaracho (Greece), Karasumi (Japan) or Bottarga (Italy), depending on the geographical production area [23,64]. Finally, the presence of bacteria producing substances with surfactant activity deriving from a brackish transitional environment deserves attention because these compounds could also have applications in bioremediation and represent an important biotechnological potential that can be furtherly investigated.

## 5. Conclusions

In conclusion, the present research has led to the selection of bacterial strains with interesting biotechnologically traits from the gut of grey mullets and has confirmed that intestinal microbiota is a promising source of new and biologically active pharmaceutical agents to control fish health and to preserve the environment. Additionally, the study of BS-producing bacteria associated with fish intestine is of relevance for our understanding of their ecological role in the symbiotic and antagonist interaction with the host and between themselves and for understanding whether the production of bioactive compounds might represent a biological strategy for protecting fish against gut and liver inflammations, as an immune response and for survival with respect to the surrounding environment.

## Figures and Tables

**Figure 1 microorganisms-09-02555-f001:**
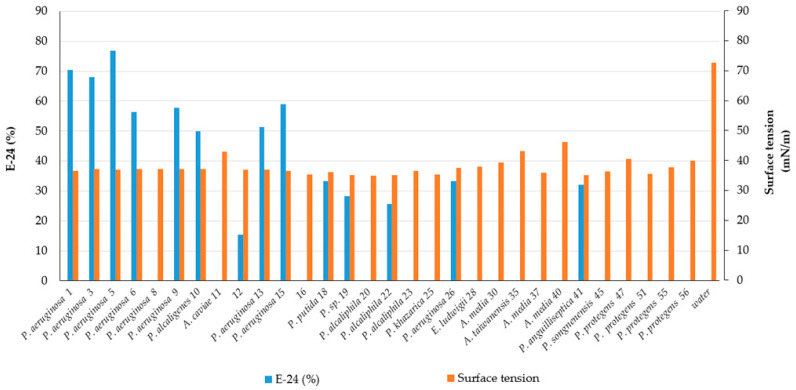
Emulsification index E-24(%) and the surface tension activity (mN·m^−1^) of intestinal bacterial cultures from mullet species.

**Figure 2 microorganisms-09-02555-f002:**
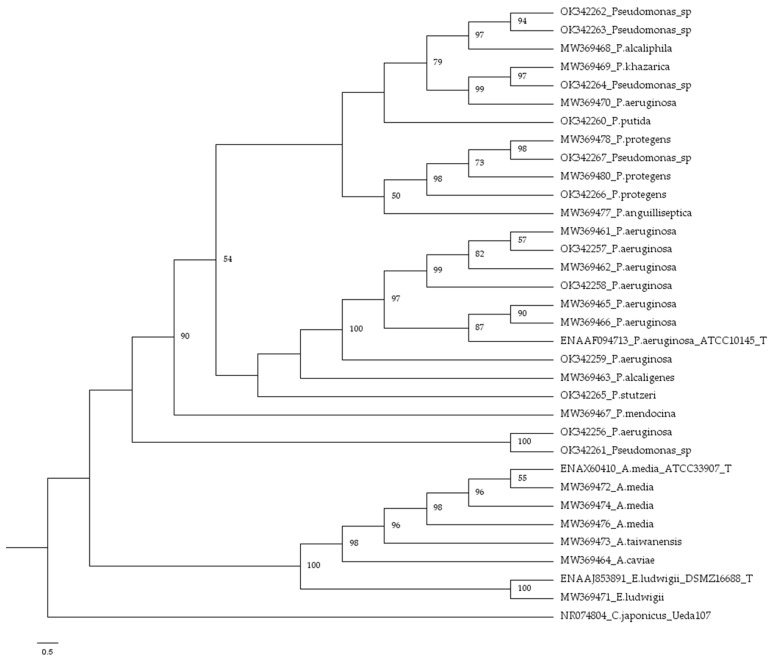
Phylogenetic tree based on 16S rRNA gene sequences comparison between the intestinal strains and reference collection strains. NR074804 *Cellvibrio japonicus* strain Ueda 107 was used as the outgroup strain. Each node indicates the percentage of the obtained bootstrap values higher than 50% of 1000 replicates. The scale bar indicates sequence divergence.

**Figure 3 microorganisms-09-02555-f003:**
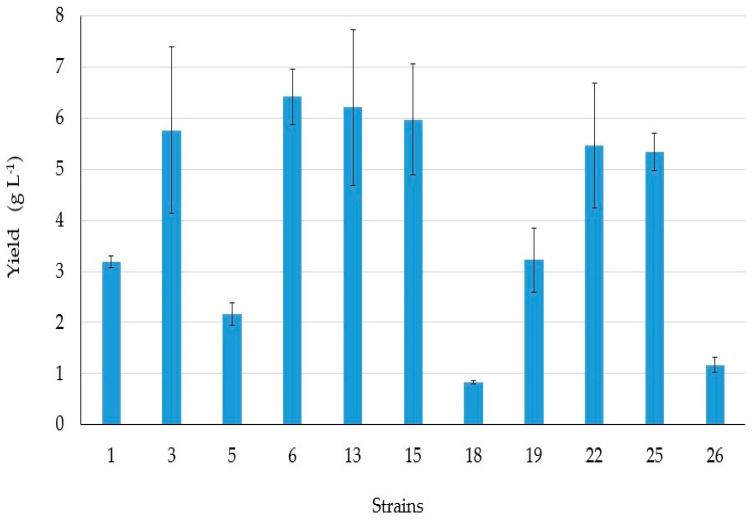
Yield of BS extracts (g L^–^^1^) from intestinal bacterial cell-free supernatants. Error bars indicate standard error (SE).

**Figure 4 microorganisms-09-02555-f004:**
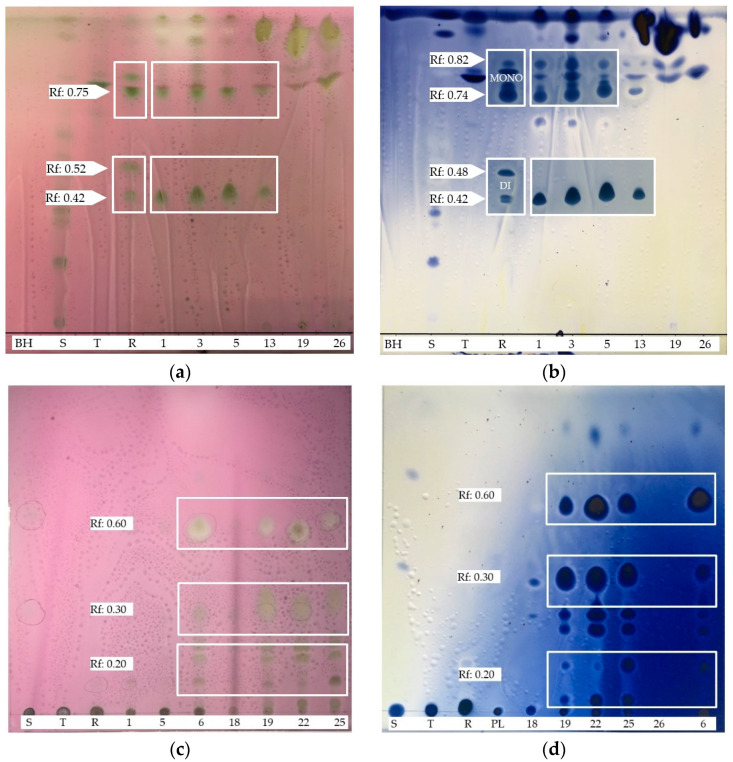
Examples of TLC plates of intestinal BS extracts stained for detecting sugars (**a**) and (**c**) and lipids (**b**) and (**d**). BH: Bushnell–Haas broth; S: sophorolipids; T: trealose lipids; R: rhamnolipids; PL: phospholipids; (**a**) and (**b**) = solvent system: chloroform: acetic acid:methanol:water (65:15:1:1); (**c**) and (**d**) = solvent system: chloroform:exane:ether:acetic acid (70:30:2).

**Figure 5 microorganisms-09-02555-f005:**
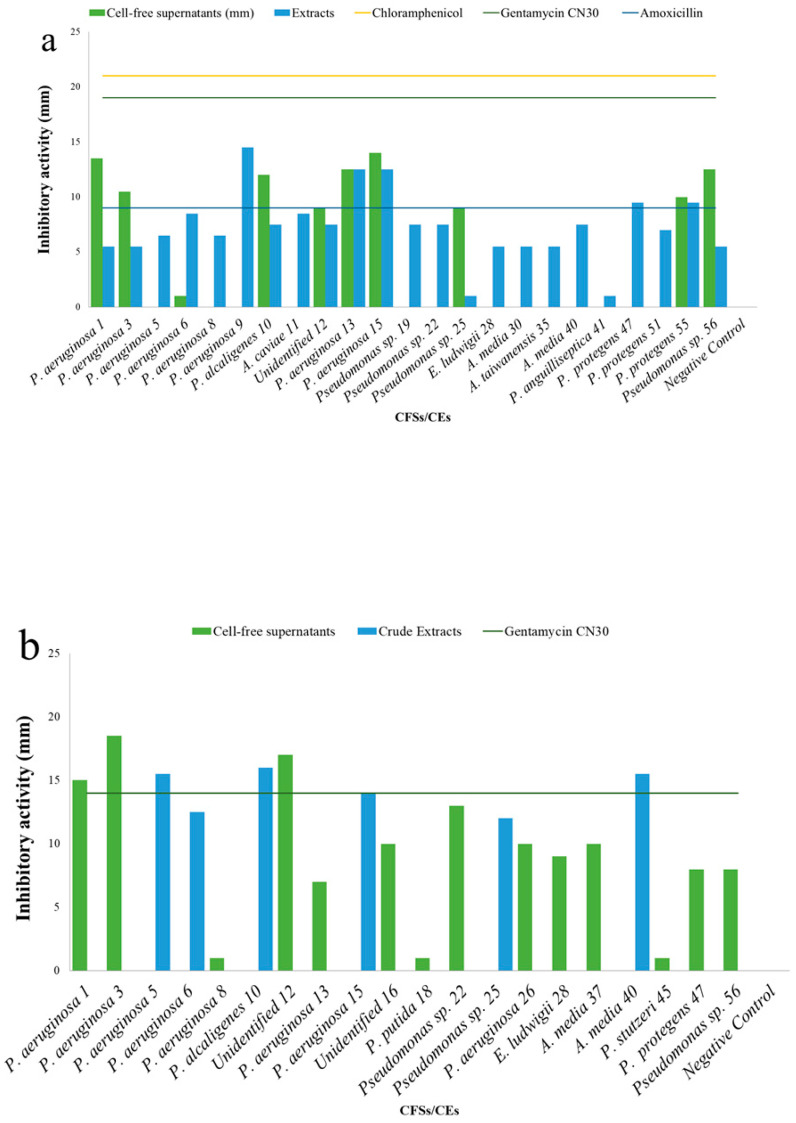
Inhibitory activity exhibited against the target *S. aureus* H1610 (**a**), *P. mirabilis* H1643 (**b**), *K. pneumoniae* H1637 (**c**) and *A. hydrophila* H1563 (**d**) by concentrated supernatants (CFSs green) and crude extracts (CEs blue) obtained by bacterial isolates.

**Table 1 microorganisms-09-02555-t001:** BS-producing bacteria from grey mullets’ guts: bacterial affiliations (similarity 99–100%), performed tests: (mean ± SD) and TLC results. Highest E24 values are highlighted in bold.

Strain	Fish Species	Bacterial Affiliation	GeneBank Accession Number	Drop Collapse	E-24 (%)	Surface TensionmN·m^−1^	BS Type
1	CR	*Pseudomonas aeruginosa*	MW369461	+++	**70.5 ± 9.1**	36.5 ± 0.1	Rhamnolipid
3	CR	*Pseudomonas aeruginosa*	OK342256	+++	**68.0 ± 12.7**	37.1 ± 0.1	Rhamnolipid
5	CR	*Pseudomonas aeruginosa*	OK342257	+++	**77.0 ± 0.0**	36.9 ± 0.4	Rhamnolipid
6	CR	*Pseudomonas aeruginosa*	MW369462	+++	**56.4 ± 0.0**	37.1 ± 0.1	Less polar compound
8	CR	*Pseudomonas aeruginosa*	OK342258	++	0.0 ± 0.0	37.1 ± 0.1	nd
9	CR	*Pseudomonas aeruginosa*	OK342259	+++	**57.7 ± 1.8**	37.2 ± 0.3	nd
10	CR	*Pseudomonas alcaligenes*	MW369463	+	**50.0 ± 1.8**	37.2 ± 0.3	nd
11	CR	*Aeromonas caviae*	MW369464	-	0.0 ± 0.0	43.0 ± 0.1	nd
12	CR	*-*	-	+++	15.4 ± 21.8	36.9 ± 0.1	nd
13	CR	*Pseudomonas aeruginosa*	MW369465	+++	**51.3 ± 3.6**	36.9 ± 0.1	Rhamnolipid
15	CR	*Pseudomonas aeruginosa*	MW369466	+++	**59.0 ± 3.6**	36.6 ± 0.6	Rhamnolipid
16	CR	*-*	-	weak	0.0 ± 0.0	35.35 ± 0.6	nd
17	CR	*Pseudomonas mendocina*	MW369467	-	20.5 ± 0	nd	nd
18	CR	*Pseudomonas putida*	OK342260	weak	**33.3 ± 3.6**	36.1 ± 0.1	Less polar compound
19	MC	*Pseudomonas* sp.	OK342261	+	**28.2 ± 3.6**	35.2 ± 0.0	Less polar compound
20	MC	*Pseudomonas alcaliphila*	MW369468	weak	0.0 ± 0.0	35.0 ± 0.4	nd
21	MC	*-*	-	weak	**25.6 ± 14.5**	nd	nd
22	MC	*Pseudomonas* sp.	OK342262	+	**25.6 ± 0.0**	35.1 ± 0.2	Less polar compound
23	MC	*Pseudomonas* sp.	OK342263	weak	0.0 ± 0.0	36.5 ± 0.1	nd
24	MC	*Pseudomonas khazarica*	MW369469	weak	0.0 ± 0.0	nd	nd
25	MC	*Pseudomonas* sp.	OK342264	+	0.0 ± 0.0	35.3 ± 0.1	Less polar compounds
26	MC	*Pseudomonas aeruginosa*	MW369470	++	**33.3 ± 0.0**	37.6 ± 0.3	Less polar compound
28	CS	*Enterobacter ludwigii*	MW369471	+	0.0 ± 0.0	37.9 ± 0.1	nd
30	CS	*Aeromonas media*	MW369472	weak	0.0 ± 0.0	39.4 ± 0.9	nd
35	CS	*Aeromonas taiwanensis*	MW369473	weak	0.0 ± 0.0	43.2 ± 0.1	nd
37	CL	*Aeromonas media*	MW369474	-	0.0 ± 0.0	35.9 ± 0.1	nd
40	CL	*Aeromonas media*	MW369476	-	0.0 ± 0.0	46.1 ± 0.3	nd
41	CL	*Pseudomonas anguilliseptica*	MW369477	+	**32.1 ± 5.4**	35.2 ± 0.6	nd
45	CL	*Pseudomonas stutzeri*	OK342265	+	0.0 ± 0.0	36.3 ± 0.1	nd
47	CL	*Pseudomonas protegens*	MW369478	weak	0.0 ± 0.0	40.5 ± 0.4	nd
51	CL	*Pseudomonas protegens*	OK342266	+	0.0 ± 0.0	35.5 ± 0.1	nd
55	CL	*Pseudomonas protegens*	MW369480	-	0.0 ± 0.0	37.7 ± 0.1	nd
56	CL	*Pseudomonas* sp.	OK342267	+	0.0 ± 0.0	39.9 ± 0.1	nd

nd: not detected; CR: Chelon ramada, MC: Mugil cephalus; CS: Chelon saliens; CL: Chelon labrosus.

**Table 2 microorganisms-09-02555-t002:** Antibacterial activity of supernatants and crude extracts (mm) in agar diffusion assay against bacterial pathogens. Values are expressed as the mean ± standard deviation of three replicates. Highest values are highlighted in bold.

	Cell-Free Supernatants (CFSs) and Crude Extracts (CEs) (mm)							
Test	*S. aureus* H1610		*P. mirabilis* H1643		*K. pneumoniae* H1637		*A. hydrophila* H1563	
	CFSs	CEs	CFSs	CEs	CFSs	CEs	CFSs	CEs
*Pseudomonas aeruginosa* 1	**13.5 ± 0.7**	5.5 ± 0.7	**15 ± 0.0**	-	-	**17.5 ± 0.7**	-	-
*Pseudomonas aeruginosa* 3	**10.5 ± 2.1**	5.5 ± 0.7	**18.5 ± 0.7**	-	-	8.0 ± 0.0	-	-
*Pseudomonas aeruginosa* 5	-	5.5 ± 0.7	-	**15.5 ± 0.7**	-	6.0 ± 0.0	-	-
*Pseudomonas aeruginosa* 6	+	8.5 ± 0.7	-	**12.5 ± 0.7**	-	**11.5 ± 0.7**	-	-
*Pseudomonas aeruginosa* 8	-	6.5 ± 0.7	+	-	-	7.0 ± 0.0	-	6.5 ± 0.7
*Pseudomonas aeruginosa* 9	-	**14.5 ± 0.7**	-	-	-	**12.0 ± 0.0**	-	-
*Pseudomonas alcaligenes* 10	**12 ± 0.0**	7.5 ± 0.7	-	**16.0 ± 0.0**	-	+	-	-
*Aeromonas caviae* 11	-	8.5 ± 0.7	-	-	-	+	-	-
Unidentified 12	9 ± 1.4	7.5 ± 0.7	**17.0 ± 1.4**	-	-	7.5 ± 0.7	-	-
*Pseudomonas aeruginosa* 13	**12.5 ± 0.7**	**12.5 ± 0.7**	7.0 ± 0.0	-	-	**12.5 ± 0.7**	-	7.0 ± 0.0
*Pseudomonas aeruginosa* 15	**14 ± 0.0**	**12.5 ± 0.7**	-	**14.0 ± 0.0**	-	**12.5 ± 0.7**	-	-
Unidentified 16	-	-	**10.0 ± 0.0**	-	-	-	-	-
*Pseudomonas putida* 18	-	-	+	-	-	-	-	-
*Pseudomonas* sp. 19	-	7.5 ± 0.7	-	-	-	8.0 ± 0.0	-	+
*Pseudomonas alcaliphila* 20	-	-	-	-	-	-	-	-
*Pseudomonas* sp. 22	-	7.5 ± 0.7	**13.0 ± 0.0**	-	-	5.5 ± 0.7	-	-
*Pseudomonas* sp. 23	-	+	+	-	-	-	-	-
*Pseudomonas* sp. 25	9 ± 0.0	+	-	**12.0 ± 0.0**	-	6.5 ± 0.7	-	-
*Pseudomonas aeruginosa* 26	-	-	**10 ± 0.0**	-	-	-	-	-
*Enterococcus ludwigii* 28	-	5.5 ± 0.7	9.0 ± 0.0	-	+	+	-	-
*Aeromonas media* 30	-	5.5 ± 0.7	-	-	-	6.0 ± 0.0	-	-
*Aeromonas taiwanensis* 35	-	5.5 ± 0.7	-	-	-	6.0 ± 0.0	-	-
*Pseudomonas protegens* 37	-	-	**10.0 ± 0.0**	-	-	-	-	-
*Aeromonas media* 40	-	7.5 ± 0.7	-	**15.5 ± 0.7**	-	+	-	+
*Pseudomonas anguilliseptica* 41	-	+	-	-	-	-	-	8.0 ± 0.0
*Pseudomonas stutzeri* 45	-	-	+	-	-	-	-	-
*Pseudomonas protegens* 47	-	9.5 ± 0.7	8.0 ± 0.0	-	-	-	-	6.5 ± 0.7
*Pseudomonas protegens* 51	-	7.0 ± 0.7	-	-	-	7.5 ± 0.7	-	-
*Pseudomonas protegens* 55	**10 ± 0.0**	9.5 ± 0.7	-	-	-	-	-	-
*Pseudomonas* sp. 56	**13.5 ± 0.7**	5.5 ± 0.7	8.0 ± 0.0	-	-	+	-	-
Negative control	0.0 ± 0.0		0.0 ± 0.0		0.0 ± 0.0		0.0 ± 0.0	
Chloramphenicol	21 ± 0.0		-		+		30.0 ± 0.0	
Gentamycin CN30	-		14		8.0 ± 0.0		18.0 ± 0.0	
Amoxycillin	-		-		-		-	

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
