# Peer review of "Isolation and Identification of Bacteria with Surface and Antibacterial Activity from the Gut of Mediterranean Grey Mullets"

_microorganisms, 2021, doi:10.3390/microorganisms9122555_

Round 1

Reviewer 1 Report

The manuscript "Intestinal microbiota of Mediterranean grey mullets: a study on the production of added-value bioactive compounds" by Floris R. et al. is devoted to the study of fish microbiome. The authors presents raw results of isolation and some assays (emulsifying, surface and antagonistic activities). Guess, that's not a finished work with clear results, as well as the presented data are hard to perception and analysis. The manuscript should be rejected in the current form.

  1. The identification and characterisation of isolated species are not fit into the current microbiological standarts. The 16S rRNA gene sequence is not enough for bacterial identification. Biochemical, morphological, cultural etc. characteristics are essential for futher analysis.
  2.  The antibiotic activity should be presented in the adequate manner: the assay is more qualitative not so precious. 11.5+/-0.7 mm - that's redundant value, the "moderate" characteristic is enough.
  3. The TLC data are not enough for compounds' identification and characterization. At least, the MS (better LC-HRMS) analysis should be perfomed for isolated material.
  4. The authors' conclusion about "bioactive compounds identified were ascribed to the class of glycolipids" is not supported by the experimental data. And it looks doubly strange when it's following the text about antibacterial activity. In general, the text is formatted in hard to read manner, especially the paragraphs with expeimental data analysis.

Author Response

Dear Reviewer 1,

the authors thank your suggestions and made many changes to the manuscript for  improving it. 

The reply to the comments is as follows:

The manuscript "Intestinal microbiota of Mediterranean grey mullets: a study on the production of added-value bioactive compounds" by Floris R. et al. is devoted to the study of fish microbiome. The authors presents (emulsifying, surface and antagonistic activities). Guess, that's not a finished work with clear results, as well as the presented data are hard to perception and analysis. The manuscript should be rejected in the current form.

  1. The identification and characterisation of isolated species are not fit into the current microbiological standarts. The 16S rRNA gene sequence is not enough for bacterial identification. Biochemical, morphological, cultural etc. characteristics are essential for further analysis.

Reply: The authors are sorry about the general comment of Referee 1, as they think that the present  results are not row and show many data. A two-year work was carried on. Surely, the title was not so centred with the paper and the article needs to be revised, therefore we are glad to improve our manuscript. We changed the title as suggested by Reviewer 2. Here a multidisciplinary research has been carried out with the aim of selecting strains that could be used for applicative purposes. Microbiological analyses, isolation and purification of bacteria were performed and biochemical tests were used to observe the growth dynamics; all tests have been done in replicates. Phylogenetic identification has been performed by 16S rRNA PCR-sequencing,  a standard reliable molecular method widely employed for identifying both cultivable and uncultivable bacteria both in the laboratory and in clinical settings [Ricardo Franco-Duarte et al., 2019 Microorganisms, 7, 130; doi:10.3390/microorganisms7050130].  According to a consistent literature and our experience, biochemical and phenotipic characters are less used for identifiying bacteria. They are often not reliable and lead to a misleading identification at species and at strain level . Moreover some characters, considered in the identification process which control metabolism products (e.g. fermentative pattern etc.) are instable and are linked to mobile genetic traits that can be lost (cured) after multiple cultivation. The artificial bacterial cultivation, can change morphological and physiological bacterial charcteristics.

  1. The antibiotic activity should be presented in the adequate manner: the assay is more qualitative not so precious. 11.5+/-0.7 mm - that's redundant value, the "moderate" characteristic is enough.

Reply: The authors retain this form more complete for presentation of results. This value represents an average value of three replicates with addition of standard deviation. The intent is to show differences occurring between strains, and between the antibacterial activity toward the different pathogens.

  1. The TLC data are not enough for compounds' identification and characterization. At least, the MS (better LC-HRMS) analysis should be perfomed for isolated material.

Reply: The TLC study was used as a qualitative preliminary analyses of the extracts in order to have some indications on the class of compounds present in the cells free supernatant. As reported in other studies, Pseudomonas genus is able to synthesize BSs of a diverse chemical nature but the more widely studied ones are low molecular weight compounds called rhamnolipids [ref. 46 Tuleva et al. 2002]. We did a preliminary biochemical characterization as Koch et al., 1991Journal of Bacteriology, 173, 4212-4219; Franzetti et. al., 2008 FEMS Microbiol. Ecol, 63, 238-248;  Samadi et al., J. Biol. Sci., 2007, 7, 1266-1269 ref.32; Aparna et al., 2012 Research in Biotechnology 3, 2:49-56, Malavenda et al. 2015 ref.15 and others. In this study we wanted to have information about the type of compounds in order to choose the best analytical conditions (column etc.) for further analyses before planning a LC-HRMS analyses. However, we used specific standards which were very useful for the classification of the presumptive glycolipids.

  1. The authors' conclusion about "bioactive compounds identified were ascribed to the class of glycolipids" is not supported by the experimental data. And it looks doubly strange when it's following the text about antibacterial activity. In general, the text is formatted in hard to read manner, especially the paragraphs with expeimental data analysis.

Reply: Thin layer chromatography is widely reported as preliminary approach for a chemical identification of biosurfactant molecules. Here, according to results obtained, we suggest a possible glycolipidic composition, as reported in the text, too. Here, the main aim was to perform a screening in order to detect possible BS-producing strains. Certainly, further analysis (object of another work) are planned for a more detailed chemical elucidation of biosurfactants, which clearly requires other more elaborate expensive procedures.

Reviewer 2 Report

REVIEW OF THE ARTICLE BY ROSANNA FLORIS ET AL. ENTITLED "INTESTINAL MICROBIOTA OF MEDITERRANEAN GREY MULLETS: A STUDY ON THE PRODUCTION OF ADDED-VALUE BIOACTIVE COMPOUNDS" (microorganisms-1479241)

The Authors isolated several strains of proteobacteria (genera Enterobacter and Pseudomonas) from the gut of the Mediterranean grey mullets (Actinopterygii, Mugelidae), i.e. Mugil cephalus, Chelon ramada, Chelon labrosus, and Chelon saliens. They studied their surfactant producing and anti-bacterial activity. In other words, they studied members of the diversity of fish gut bacteria by conventional microbiological methods. Although current knowledge about gut microbiome is obtained mainly from NGS-based research, support of these data by means of classical microbiology and study of bacterial strains is vitally important. Therefore the data reported are valuable. Introduction is satisfied, surfactant production and antibacterial activity are adequately measured, identification of bacteria is substandard and must be improved. In general, Discussion is sufficient, but some issues should be removed. Therefore, I suggest major revision of the text in accordance with the list of criticisms (see below).

MAJOR CRITICISMS

-Title must be changed. It does not reflect the work. The Authors did not study diversity of fish gut microbes, but selected the strains with the phenotype of interest, they did not study the repertoire of value-added compounds, surfactant activity only. It would be e.g. “Isolation and identification of bacteria with surfactant-producing and antibacterial activity from the gut of the Mediterranean grey mullets”.

-The procedure of bacterial strains identification is unsatisfactory. It cannot be done by a simple blast search. To determine taxonomic affinity, phylogenetic tree reconstruction is required. 1) The Authors should select verified sequences of the taxa at the genus and species level including mandatory type strains (it can be found e.g. on https://bacdive.dsmz.de/), an outgroup taxon should be indicated. 2) Multiple alignment should be constructed by an published algorithm with corresponding reference. 3) Phylogeny analysis should be done. Preferably a cladistic algorithm such as Maximum Likelihood (Felsenstein, J. (1981) Journal of molecular evolution 17(6): 368-376) or Bayesian Inference (Altekar et al. (2004) Bioinformatics 20(3): 407-415). 4) In the first case, topology robustness tests, such as bootstrap (Felsenstein (1985) evolution 39(4): 783-791), and selection of the best DNA evolution model have to be performed. In the second case statistical significance of nodes should be verified by posterior probabilities. The results must be presented as it is commonly accepted in recent works.

-Since fish is a higher animal, according to standards of scientific publication, You must indicate the corresponding ethical statement. The procedure of fish euthanasia must be performded and presented in the text in according with current standards for this group of animals (see and cite the guidelines Underwood, Wendy, et al. "AVMA guidelines for the euthanasia of animals: 2013 edition." Schaumburg, IL: American Veterinary Medical Association, 2013).

-The Authors use the terms ‘strain’ and ‘isolate’ through the text. Please, explain the difference.

-l. 26-28. Please, provide units.

-l. 20. By definition, surfactants are compounds with surface activity (antibacterial activity is not a mandatory feature). Please, modulate.

-l. 185. The researchers should not edit obtained sequences! They can only remove initial and final noize positions. Edited sequences cannot be a source of reliable information.

-l. 189. NCBI is not a database. It is National Center for Biotechnology Information. Most likely, You used the NCBI GenBank nucleotide database. Did You set up your filters to uncultured and environmental samples?

-l. 196-197. Provide names of the strains.

-Table 1. Please, explain bolded and non-bolded fonts, explain the scores +, ++, +++. Each table should be self-explaining.

-. Table 1. ‘number’ should be GenBank ID or GenBank accession number.

-Table 1. Please, explain, what is type.

-Figure 2. Capture is too brief. Extracts from what? What is BS? What do values and error bars mean?

-More appropriate term for Rf is retardation factor rather than retention factor (see IUPAC guidelines).

-l. 298-310. It is better to replace “spots” to “TLC fractions”. How did You determine the retardation factor for spots with non-zero size? Please, explain in the text.

-l. 303. “These defined compounds co-band with the rhamnolipids standard” rephrase to “were characterized by the same Rf values as the rhamnolipid standard”.

-In antibacterial activity test the data on negative controls must be on the figure 4 and Table 2.

-Figure 4. Describe CFSs and CEs in legend. K. pneumoniae should be italicized. Indicate names of the strains, not species only.

-Table 2. Please, explain bolded and non-bolded fonts. Each table should be self-explaining. Indicate names of the strains, not species only.

-l. 366. “s like biosurfactants with surface and antimicrobial” - there is no experimental evidence for the conclusion that the same molecules were responsible for surface activity and antibacterial activity. Did You analyze the presence of potential antibacterial substances in these extracts? TLC data are insufficient.

-l. 368-370. There is no experimental evidence for this statement.

MINOR CRITICISMS

-l. 34-36. The ref. [1] of 2012, in my opinion, does not fit the statement “s has recently become a precious tool ”.

-L. 65-66. “by [18]” should be  “Giri et al. [18]”, “by [19]” should be “by “Mukherjee et al. [19]”. In addition, check the format of authors’ names in [18].

-l. 79. Remove Durad et al. 2013.

-l. 98-99. What is d?

-In the introduction and discussion you can also discuss isolation and production of exopolymeric substances by fish gut bacteria according to resently published works, eg 10.1007/s00248-020-01652-0.

-l. 132. “HCl 1N” nust be “1N HCl”.

-l. 159. Provide manufacturer for silica gel plates.

-l. 160. 166. For comprehension, in composition of solvents or dyes systems commas should be changed to “:”. For example, “chloform, methanol, acetic acid and water” should be “chloroform:methanol:acetic acid:water”.

-l. 161. Remove one “/”

-l. 165-170. Both manufacturers and countries of origin and names of standards must be provided for each reagent.

-l. 177. Provide reference for primers.

-l. 187. Reference should be added to the list and presented in the correct format.

-l. 194, 359. Do You really mean pathogenic bacteria? “Bacterial pathogens” is more appropriate to cause diseases of bacteria.

-l. 217, 218. Names of reagents cannot be with a capital letter.

-l. 240. Comma is missing.

-l. 300-301. Panels should be ordered on their mention in the text.

-l. 423. Insoluble where? In water?

-l. 459. It is a very debatable statement. There are many such data, see, e.g. recent reviews, eg Egerton et al (2018) Frontiers in microbiology, 9, 873, Gallo et al. (2020) Fisheries, 45(5), 271-282, Butt, Volkoff (2019) Frontiers in endocrinology, 10, 9.etc.

Author Response

Dear Referee 2,

the authors would like to thank you for your precious and detailed observations to improve the manuscript. The replies to all the points are as follows:

The Authors isolated several strains of proteobacteria (genera Enterobacter and Pseudomonas) from the gut of the Mediterranean grey mullets (Actinopterygii, Mugelidae), i.e. Mugil cephalusChelon ramadaChelon labrosus, and Chelon saliens. They studied their surfactant producing and anti-bacterial activity. In other words, they studied members of the diversity of fish gut bacteria by conventional microbiological methods. Although current knowledge about gut microbiome is obtained mainly from NGS-based research, support of these data by means of classical microbiology and study of bacterial strains is vitally important. Therefore the data reported are valuable. Introduction is satisfied, surfactant production and antibacterial activity are adequately measured, identification of bacteria is substandard and must be improved. In general, Discussion is sufficient, but some issues should be removed. Therefore, I suggest major revision of the text in accordance with the list of criticisms (see below).

MAJOR CRITICISMS

-Title must be changed. It does not reflect the work. The Authors did not study diversity of fish gut microbes, but selected the strains with the phenotype of interest, they did not study the repertoire of value-added compounds, surfactant activity only. It would be e.g. “Isolation and identification of bacteria with surfactant-producing and antibacterial activity from the gut of the Mediterranean grey mullets”.

Reply: The Reviewer 2 is right.The title was quite a bit general. We changed it as suggested.

-The procedure of bacterial strains identification is unsatisfactory. It cannot be done by a simple blast search. To determine taxonomic affinity, phylogenetic tree reconstruction is required. 1) The Authors should select verified sequences of the taxa at the genus and species level including mandatory type strains (it can be found e.g. on https://bacdive.dsmz.de/), an outgroup taxon should be indicated. 2) Multiple alignment should be constructed by an published algorithm with corresponding reference. 3) Phylogeny analysis should be done. Preferably a cladistic algorithm such as Maximum Likelihood (Felsenstein, J. (1981) Journal of molecular evolution 17(6): 368-376) or Bayesian Inference (Altekar et al. (2004) Bioinformatics 20(3): 407-415). 4) In the first case, topology robustness tests, such as bootstrap

 (Felsenstein (1985) evolution 39(4): 783-791), and selection of the best DNA evolution model have to be performed. In the second case statistical significance of nodes should be verified by posterior probabilities. The results must be presented as it is commonly accepted in recent works.

Reply: The authors, following the indications of the referee  2 , did a phylogenetic tree using the 16S ribosomal gene bacterial sequences obtained in the study with the reference strains ENA AF094713 P. aeruginosa strain ATCC 10145, ENA X60410 Aeromonas media strain ATCC 339007, ENA AJ853891 Enterobacter ludwigii and the strain NR074804 Cellvibrio japonicum STRAIN Ueda 107 was used as the outgroup,  according to Gomila et al.2015 Front. Microbiol. 6, Phylogenomics and systematics in Pseudomonas, https://doi.org/10.3389/fmicb.2015.00214 ref.36. MEGA x (Kumar et al. 2018 doi:10.1093/molbev/msy096 ref.37) was used to align the 8200bp 16S ribosomal gene sequences. A phylogenetic tree was reconstructed by Clustal W Multiple alignment and the phylogenetic tree was inferred through Maximum Likelihood (ML) (Kalyaanamoorthy et al. 2017 ref.38). The ML phylogenetic tree was calculated under the best model K2P+G4 ref. 39 with IQTREE webserver ref.40. Clade support was estimated through 1000 ultrafast bootstrap replicates. We inserted all these information in in the text  (methods Lines 321-328), and we inserted a new Figure 2 in the results.

-Since fish is a higher animal, according to standards of scientific publication, You must indicate the corresponding ethical statement. The procedure of fish euthanasia must be performed and presented in the text in according with current standards for this group of animals (see and cite the guidelines Underwood, Wendy, et al. "AVMA guidelines for the euthanasia of animals: 2013 edition." Schaumburg, IL: American Veterinary Medical Association, 2013).

Reply: As we told inside the text we used dead fish destined to the food market. We did not sacrify any animals. The authors explained better the samplings (Line 104-109)

-The Authors use the terms ‘strain’ and ‘isolate’ through the text. Please, explain the difference.

Reply: Isolate is used in the first phase of the work when we isolated the colony from the agar plate. Then the isolates were purified.The strains are the purified colony after different (three) streakes on a solidified agar medium to obtain a pure culture.

-l. 26-28. Please, provide units.

Reply: we insert  the units (mm)

-l. 20. By definition, surfactants are compounds with surface activity (antibacterial activity is not a mandatory feature). Please, modulate.

Reply: Our study is focused on biosurfactants which are widely studied for surface and antibacterial activities and we did the same.

-l. 185. The researchers should not edit obtained sequences! They can only remove initial and final noize positions. Edited sequences cannot be a source of reliable information.

Reply: The authors just removed initial and final noize positions we used the term edit not properly, sorry. We modified the text according to what we did.

 -l. 189. NCBI is not a database. It is National Center for Biotechnology Information. Most likely, You used the NCBI GenBank nucleotide database. Did You set up your filters to uncultured and environmental samples?

Reply: Yes it is. I changed the text as indicated. The authors did not consider the ulculturable and environmental samples .

-l. 196-197. Provide names of the strains.

Reply: We provided the names of the strains.

-Table 1. Please, explain bolded and non-bolded fonts, explain the scores +, ++, +++. Each table should be self-explaining.

Reply: I explained the bolded font in the Table caption.The bolded font evidences the highest E-24 values.

-. Table 1. ‘number’ should be GenBank ID or GenBank accession number.

-Table 1. Please, explain, what is type

Reply: Accession number. Submission SUB10458851.  I explained it.

-Figure 2. Capture is too brief. Extracts from what? What is BS? What do values and error bars mean?

Reply: Yes, the authors enlarged the capture. Figure 2. Yield of BS extracts (g L-1) from intestinal bacterial cell-free supernatants. Error bars indicates standard error (se) .

-More appropriate term for Rf is retardation factor rather than retention factor (see IUPAC guidelines).

Reply: We changed retention factor into retardation factor.

-l. 298-310. It is better to replace “spots” to “TLC fractions”. How did You determine the retardation factor for spots with non-zero size? Please, explain in the text.

Reply: we replaced the term “spots” to TLC fractions. The retardation factor (Rf) was calculated in the conventional way and explained it in the text. Each retardation factor (Rf) was calculated dividing the distance of the considered TLC fraction run in the TLC plate from the origin by the distance of the solvent from the same origin. We added it in the materials and methods.

-l. 303. “These defined compounds co-band with the rhamnolipids standard” rephrase to “were characterized by the same Rf values as the rhamnolipid standard”.

-In antibacterial activity test the data on negative controls must be on the figure 4 and Table 2.

Reply: Figure 5 and Table 2 have been revised according to the Reviewer 2.

-Figure 4. Describe CFSs and CEs in legend. K. pneumoniae should be italicized. Indicate names of the strains, not species only.

Reply: Figure 4 has became Figure 5 has been revised accordingly.

-Table 2. Please, explain bolded and non-bolded fonts. Each table should be self-explaining. Indicate names of the strains, not species only.

Reply: I explained the Table caption. The bolded font evidences the most active strains.

-l. 366. “s like biosurfactants with surface and antimicrobial” - there is no experimental evidence for the conclusion that the same molecules were responsible for surface activity and antibacterial activity. Did You analyze the presence of potential antibacterial substances in these extracts? TLC data are insufficient.

Reply: Surface tension measurements have been performed on cell-free supernatants, the same tested for antibacterial assays and the same from which biosurfactants have been extracted to perform antibacterial assays. The surface and antibacterial activities are ascribed to the same type of molecules by the literature [Vatsa et. al., 2010 Rhamnolipid <biosurfactants as New Players in Animal and plant Defense against Microbes. Int. J. Mol. Sci. 11,12, 5095-5108 and others ].

-l. 368-370. There is no experimental evidence for this statement.

Reply: The emulsifying activities and the reduction of surface tension are considered good indicator of biosurfactant production (Twigg et al., 2020Microbial Biotechnology (2020), doi:10.1111/1751-7915.13704 ref. 43). On the base of this assessment, we suggest biosurfactant production by the tested strains. The production of BSs is influenced by the growth conditions as well documented .[Samadi et al., 2007 ref.32; Rizzo et al., Environ Sci Pollut Res (2014), DOI 10.1007/s11356-013-2259-8 [ref. 44]; Rizzo et al. J Basic Microbiol. 2018; DOI: 10.1002/jobm.201700669 [ ref. 45];

MINOR CRITICISMS

-l. 34-36. The ref. [1] of 2012, in my opinion, does not fit the statement “s has recently become a precious tool ”.

Reply: The authors intended to emphasize the importance of approach in the bioprospecting field. The sentence has been remodulated.

-L. 65-66. “by [18]” should be  “Giri et al. [18]” Line 81, “by [19]” should be “by “Mukherjee et al. [19]”Line 83. In addition, check the format of authors’ names in [18].

Reply: Corrected in the text.

-l. 79. Remove Durad et al. 2013.

Reply: Corrected in the text Line 96.

-l. 98-99. What is d?

Reply: d indicates the preciousness of the measurement accuracy

-In the introduction and discussion you can also discuss isolation and production of exopolymeric substances by fish gut bacteria according to resently published works, eg 10.1007/s00248-020-01652-0.

Reply: Thank you for the suggestions. The reference 14 has been added, and discussed now inserted Introduction ( Line 72) and Discussion (Lines 563)

-l. 132. “HCl 1N” nust be “1N HCl”.

Reply: Line 153: We corrected to  1N HCl

-l. 159. Provide manufacturer for silica gel plates.

Reply: Provided in the text ©Millipore Corporation.

-l. 160. 166. For comprehension, in composition of solvents or dyes systems commas should be changed to “:”. For example, “chloform, methanol, acetic acid and water” should be “chloroform:methanol:acetic acid:water”.

Reply: corrected

-l. 161. Remove one “/”

Reply: removed

-l. 165-170. Both manufacturers and countries of origin and names of standards must be provided for each reagent.

Reply: Dr. Johannes Kugler from Karlsruhe Institute of Technology  (Germany) provided the standards sophorolipids and trealose lipids while the mix of rhamnolipids were from Sigma Aldrich and phospholipids (Supelco) (Lines 235-238). We changed the acronym in the text for a better correspondence to the Figure 4 (Lines 234-235).

-l. 177. Provide reference for primers

Reply: Marchesi et al., 1998 inserted in the literature ref 34.

-l. 187. Reference should be added to the list and presented in the correct format.

Reply: Added in the list ref 35.

-l. 194, 359. Do You really mean pathogenic bacteria? “Bacterial pathogens” is more appropriate to cause diseases of bacteria.

Reply: Changed pathogenic bacteria to bacterial pathogens in the text Lane 332.

-l. 217, 218. Names of reagents cannot be with a capital letter.

Reply: Changed in the text.

-l. 240. Comma is missing.

Reply: full stop is missing?

-l. 300-301. Panels should be ordered on their mention in the text.

Reply: The authors preferred to show first the plates (a and b) which present the same samples and run using the first type of solvent and successively the plates (c and d) run with the second type of solvent.

-l. 423. Insoluble where? In water?

Reply: Yes Insoluble in water. Rhamnolipids have the characteristic to solubilise hydrocarbon which are insoluble in water.

-l. 459. It is a very debatable statement. There are many such data, see, e.g. recent reviews, eg Egerton et al (2018) Frontiers in microbiology9, 873, Gallo et al. (2020) Fisheries45(5), 271-282, Butt, Volkoff (2019) Frontiers in endocrinology10, 9.etc.

Reply: Yes it is a very debatable statement. We changed the sentence in the text (Line 702). Gallo et al. 2020 state that investigations into the gut microbioma of fish have only recently emerged in the field of fisheries science  and this was what the authors did mean.

Round 2

Reviewer 1 Report

  1. Ok, some biochemical characteristics are variable, but there are essential (housekeeping) genes. And authours should perform the multi-locus secuencing (using actual housekeeping genes for each taxon based on 16S rRNA genes) to improve strain identification. That's the minimal requiring data for reliable identification.
  2. The antibiotic activity should be presented in the adequate manner: the assay is more qualitative not so precious. 11.5+/-0.7 mm - that's redundant value, the "moderate" characteristic is enough.
  3. The antibiotic activity assay is not linked with all other material and should be excluded or extended.
  4. To improve the chemical identification of biosurfactants the additional MS analysis should be performed.

Author Response

Dear Reviewer 1,

the Authors thank you for all your suggestions. Herein their reply:

  1. Ok, some biochemical characteristics are variable, but there are essential (housekeeping) genes. And authours should perform the multi-locus secuencing (using actual housekeeping genes for each taxon based on 16S rRNA genes) to improve strain identification. That's the minimal requiring data for reliable identification.

Reply: The authors considered the 16S rRNA gene sequence a sufficient and reliable method for identifying the bacteria in the present work. Indeed the purpose of their study was to isolate and identify biosurfactant producing strains from the intestinal tract of fish for applicative purposes. The multilocus sequence analysis (MLSA), which consider other house keeping genes, can be a subsequent method to improve the identification of these strains for other phylogenomics and systematics studies. The authors commented this aspect in the discussion (lines 844-848).

2. The antibiotic activity should be presented in the adequate manner: the assay is more qualitative not so precious. 11.5+/-0.7 mm - that's redundant value, the "moderate" characteristic is enough.

Reply: The authors agree that the test is mainly qualitative, but prefer to furnish all details of the experiments, as results of three replicates.

3.The antibiotic activity assay is not linked with all other material and should be excluded or extended.

Reply: The authors think that the antibiotic activity is complementary to all the other analyses performed in the study in fact the same cell free supernatants, coming from the same bacterial culture, were tested both for surface activity assays (emulsifying activities and surface tension), and for antibiotic activity. The authors retain that this has an important scientific value because it confirms what the literature states about the numerous biosurfactants proprieties (surface and antimicrobial activities etc.); moreover, all this is interesting for an applicative purpose of these bacterial products because the studied intestinal bacterial supernatants represent a natural resource of fish origin that can be experimented in aquaculture as probiotics.

4. To improve the chemical identification of biosurfactants the additional MS analysis should be performed.

Reply: As already said, further analyses (object of another work) are planned for improving the chemical identification of the biosurfactants.

Reviewer 2 Report

The Authors have signifficantly improved the work. Most issues have been removed. However there are still some minor points in he text.

  1. The procedure of phyclogenetc analysis was added but it is still not clear desribed (l. 273-279). You cannot obtain Clustal W multiple alignment by the ML algorithm, but the multiple alignment is used for the ML nanalysis. Plese, provide the references for the ML analysis and bootstrap mehod (I have provied them in the 1st report). Moreover there is no 16S ribosomal gene, 16S ribosomal RNA gene only. Most likely You meant 

"A phylogenetic tree was recon structed using the 16S ribosomal RNA gene sequences obtained in the study and the reference strains ENA AF094713 P. aeruginosa ATCC 10145, ENA X60410 Aeromonas media ATCC
275 339007, ENA AJ853891 Enterobacter ludwigii. As an out-group, the strain NR074804 Cellvibrio japonicum strain Ueda was utilised for the analysis [36]. A Clustal W Multiple
alignment was obtained by MEGA X [37] with default parametrs. Maximun Likelihood tree [38]; the
278 bootstrap test of 1000 replications was performed. The final dataset was constructed included using 820 bp positions sequences. The best fit DNA evolution model selected on the  IQTREE webserver [40] was K2P G4 [39], by IQTREE webserver [40]. Phylogenetic tree was inferred by the Maximum Likelihood algorithm [REFERENCE] in MEAGA X with default parameters. The bootstrap test (1,000 replicates) [reference] was used to evaluate robustness of tree topology."

2. l. 284-285. You still have not indicate names of used strains, species names only.

3. Please, highkight Your strains and indicate tyoe strains by T on fig. 2. In the caption: explane, please, that pecentages of bootstrap support are shown near each node, indicate what does scale bar mean.

4. I am not sure that the tree (fig 2) was obtained by the ML. Most likelu its is UPGMA tree, which is not applicable for the analysis. Please, use the option Maximum Likelihood e.g. in MEGA X.

5. For comprehension, please, use full names of genera in Table 2, Each table should be self-explaining.

Author Response

Dear Reviewer 2,

the Authors thank you for your suggestions. Herein, their reply:

Authors have signifficantly improved the work. Most issues have been removed. However there are still some minor points in he text.

  1. The procedure of phyclogenetc analysis was added but it is still not clear desribed (l. 273-279). You cannot obtain Clustal W multiple alignment by the ML algorithm, but the multiple alignment is used for the ML nanalysis. Plese, provide the references for the ML analysis and bootstrap mehod (I have provied them in the 1st report). Moreover there is no 16S ribosomal gene, 16S ribosomal RNA gene only. Most likely You meant 

Reply: it was not properly written, sorry, but the analysis was made as suggested. We corrected the text in the materials and methods lines 327-335.

"A phylogenetic tree was recon structed using the 16S ribosomal RNA gene sequences obtained in the study and the reference strains ENA AF094713 P. aeruginosa ATCC 10145T, ENA X60410 Aeromonas media ATCC275 339007T, ENA AJ853891T Enterobacter ludwigii. As an out-group, the strain NR074804 Cellvibrio japonicum strain Ueda was utilised for the analysis [36]. A Clustal W Multiple
alignment was obtained by MEGA X [37] with default parametrs. Maximun Likelihood tree [38]; the
278 bootstrap test of 1000 replications was performed. The final dataset was constructed included using 820 bp positions sequences. The best fit DNA evolution model selected on the  IQTREE webserver [40] was K2P G4 [39], by IQTREE webserver [40]Phylogenetic tree was inferred by the Maximum Likelihood algorithm [REFERENCE] in MEAGA X with default parameters. The bootstrap test (1,000 replicates) [reference] was used to evaluate robustness of tree topology.

Reply: The authors thank you for the suggestions that helped them to improve significantly the manuscript. The procedure was not properly explained. The authors rewrite what they did in the text of the materials and methods lines 327-335 and added the ref. 40 (Felsenstein et al., 1981 for Maximum Likelihood) and ref. 42 (Felsenstein et al., 1985 for topology robustness tests) according to referee 2.

  1. 284-285. You stiedll have not indicate names of used strains, species names only.

Reply: The names of used strains have been added as suggested.

  1. Please, highkight Your strains and indicate tyoe strains by T on fig. 2. In the caption: explane, please, that pecentages of bootstrap support are shown near each node, indicate what does scale bar mean.

Reply: The authors highlighted the strains, indicated the type strains by T in Figure 2, and which percentages of bootstrap support are shown near each node and what does scale bar mean in the caption. A new Figure 2 was inserted in the results.

  1. I am not sure that the tree (fig 2) was obtained by the ML. Most likelu its is UPGMA tree, which is not applicable for the analysis. Please, use the option Maximum Likelihood e.g. in MEGA X.

Reply: The authors are sure that the tree was obtained by the ML and not UPGMA. They used the option Maximum Likelihood in IQTREE and not in MEGA. The visualization of the tree was obtained by Figtree software (version 1.4.4) http://tree.bio.ed.ac.uk/software/figtree. They inserted a new Figure 2.

  1. For comprehension, please, use full names of genera in Table 2, Each table should be self-explaining.

Reply:. The authors added the full name of genera